# Animal-Origin Prebiotics Based on Chitin: An Alternative for the Future? A Critical Review

**DOI:** 10.3390/foods9060782

**Published:** 2020-06-12

**Authors:** Aroa Lopez-Santamarina, Alicia del Carmen Mondragon, Alexandre Lamas, Jose Manuel Miranda, Carlos Manuel Franco, Alberto Cepeda

**Affiliations:** Laboratorio de Higiene Inspección y Control de Alimentos. Departamento de Química Analítica, Nutrición y Bromatología, Facultad de Veterinaria, Universidade de Santiago de Compostela, 27002 Lugo, Spain; aroa.lopez.santamarina@rai.usc.es (A.L.-S.); aliciamondragon@yahoo.com (A.d.C.M.); alexandre.lamas@usc.es (A.L.); josemanuel.miranda@usc.es (J.M.M.); carlos.franco@usc.es (C.M.F.)

**Keywords:** chitin, chitosan, prebiotic, polysaccharides, gut microbiota, insect, crustacean

## Abstract

The human gut microbiota has been revealed in recent years as a factor that plays a decisive role in the maintenance of human health, as well as in the development of many non-communicable diseases. This microbiota can be modulated by various dietary factors, among which complex carbohydrates have a great influence. Although most complex carbohydrates included in the human diet come from vegetables, there are also options to include complex carbohydrates from non-vegetable sources, such as chitin and its derivatives. Chitin, and its derivatives such as chitosan can be obtained from non-vegetable sources, the best being insects, crustacean exoskeletons and fungi. The present review offers a broad perspective of the current knowledge surrounding the impacts of chitin and its derived polysaccharides on the human gut microbiota and the profound need for more in-depth investigations into this topic. Overall, the effects of whole insects or meal on the gut microbiota have contradictory results, possibly due to their high protein content. Better results are obtained for the case of chitin derivatives, regarding both metabolic effects and effects on the gut microbiota composition.

## 1. Introduction

According to United Nations predictions, it is expected that 9.8 billion people will inhabit the planet by around 2050 and 11.2 billion in 2100 [1]. Consequently, it is expected that the demand for food will increase by about 60% in the coming decades [2,3]. Thus, it will be a great challenge to ensure enough and safe food for such a large population. Also, the traditional methods usually employed to obtain food, of both of plant and animal origin, present specific problems, such as high CO_2_ emissions or a need for cultivable surfaces, that make difficult their use to meet such an ambitious increase target [4].

Thus, conventional agriculture presents concerns. For example, fresh water (an essential resource) is a very scarce commodity, and progressive desertification of the earth’s surface is taking place and will probably be aggravated in the future by global warming and desertification process [5]. According to the Food and Agriculture Organization of the United Nations (FAO) [6], it is expected a progressive decline in global agricultural productivity of about 1% per year.

With respect to animal-origin foods, their global demand is increasing drastically, and it is expected to increase even more in the future [7]. The production of animal-origin food also presents specific challenges, such as the fact that intensive production methods require a large amount of land, water and feed, and some animals (such as ruminants) produce large amounts of greenhouse gas emissions. Alternative animal protein substitutes must therefore be adopted to overcome this problem [4]. This means that the demand for food produced from non-traditional sources is expected to increase in the next decade [2].

Additionally, nowadays in most Western countries there is a growing prevalence of diet-related chronic diseases [8]. As a result, consumers, aware of this problem, demand foods with a nutritional composition more according to the recommendations of the health authorities [8,9]. In addition, the consumption of foods containing bioactive components, or foods reformulated to reduce the content of components with harmful effects on human health, has also been increasing [10]. One of the food components that are usually ingested in a lesser proportion than recommended in Western societies is dietary fiber (DF) [8,11]. Among the beneficial effects of DF on human health, a broad spectrum has been reported, such as prevention of carcinogenesis, cardiovascular diseases, metabolic syndrome, type-2 diabetes and obesity [12]. In particular, the consumption of adequate amounts and types of DF has been extensively investigated in recent years because of their effect on the gut microbiota (GM) as prebiotics [10]. The FAO (2006) defines prebiotics as “undigestible food ingredients that beneficially affect the host by selectively stimulating the growth and/or activity of one or a limited number of bacterial species that are already established in the colon and therefore improve the health of the host” [13]. Several components of DF recognized as prebiotics [5] and used as functional ingredients in the food industry [14], with the aim to act as substrates that are used selectively by host microorganisms that confer a health benefit [15].

Some DF types affect the digestion rate by reducing gastric emptying, limiting digestive enzyme activity and restricting the rate and extent of nutrient absorption in the gut [11]. Food products that naturally contain DF, such as cereals, fruits, vegetables, nuts and beans, are the main sources of DF intake [11]. Some non-vegetable foods can be also a source of DF. However, consumers are often reluctant to change their nutritional patterns to include more vegetable foods in their diet [8]. Thus, although tunicates are the only animal group known to synthesize cellulose, the most abundant natural compound in the world [16], animals and molds can produce other DF such as chitin. Chitin is a natural polysaccharide considered to be one of the most abundant biopolymers in nature [7]; is estimated to be the second most abundant biomass in the world after cellulose and forms an important structural component of many organisms, including fungi, crustaceans, mollusks, coelomates, protozoa and green algae [17].

However, the human genome encodes a short number of hydrolases capable of hydrolyzing the glycosidic bonds of polysaccharides in DF (collectively referred to as CAZymes) [18]. Through a long period of co-evolution between GM and host, intestinal microbes have evolved diverse strategies for degrading polysaccharides from terrestrial vegetables [19]. However, because the consumption of animal-origin polysaccharides such as chitin by humans was not common in most geographical areas, the human GM did not acquire the same efficacy to degrade these polysaccharides. In contrast, the gut microbiome codes tens of thousands of CAZymes that could act as a reservoir of CAZymes that could transfer them to host [19]. In this sense, it was previously demonstrated that specific genes coding for CAZymes, such as porphyranases and agarases, can be transferred from a member of marine bacteria to GM bacteria [20].

The main objective of this review is to provide an overview of the latest scientific evidence on the effect of animal-derived DF on the human GM. In addition, information will be collected about the potential to use chitin and its derivatives as prebiotics for maintaining human health.

## 2. Influence of the Gut Microbiota on Human Health

The human gastrointestinal tract harbors 10–100 times the number of total eukaryotic cells of the human body. Their counts are especially high in the distal part of the colon, reaching 10^11^–10^12^ bacteria per gram [11]. Consequently, distal part of the colon contents one of the highest densities of bacteria on the earth [11]. The dominant gut microbial phyla are Firmicutes, Bacteroidetes, Actinobacteria, Proteobacteria, Fusobacteria and Verrucomicrobia but the phyla Firmicutes and Bacteroidetes [20] represent about 90% of the total GM. The phylum Firmicutes includes more than 200 different genera, *Clostridium* being the most relevant as it represents about 95% of the Firmicutes phylum. Other important Firmicutes genera are *Lactobacillus*, *Bacillus*, *Enterococcus* and *Ruminococcus* [20]. Regarding the phylum Bacteroidetes, the predominant genus is *Bacteroides* (about 90% of total Bacteroidetes), *Prevotella* also being a relevant genus [5]. The phylum Actinobacteria is proportionally less abundant and is mainly represented by the genus *Bifidobacterium* [8]. Proteobacteria includes most food-borne bacteria, such as *Salmonella*, *Klebsiella*, *Yersinia* and *Escherichia* [21], whereas Fusobacteria is usually related to negative effects as it includes some genera such as *Fusobacterium* that are related to the development of colorectal cancer [5]. Finally, the phylum Verrucomicrobia includes the genus *Akkermansia*, a mucin-degrading bacterium believed to contribute to intestinal health and glucose homeostasis, that although is often present in small amounts has important functions and its presence is associated with good gut health [22].

The optimal GM is not yet defined as it presents great variability between individuals [4], but there is a broad consensus that a rich and diverse microbial community leads to a well-balanced and healthy GM composition [23]. Indeed, the human GM is characterized by an inter-individual variability due to various factors relating to the subject’s life history, such as infant transitions and antibiotic use, as well as lifestyle, dietary and cultural habits [24].

In recent years, a large amount of scientific literature has been published that shows a close relation between GM composition and functionality and numerous non-transmissible diseases, such as cardiovascular diseases [25], obesity [8], diabetes [26], cancer [27], gastrointestinal diseases [28] and neurological disorders [29]. Although nowadays it is not entirely known if changes in GM composition are a cause or consequence of a given disease, a positive association between the richness and diversity of the GM and human health has been demonstrated [23].

Diet is one of the key modulators of GM composition that directly affects host homeostasis and biological processes but also via metabolites derived from the microbial fermentation of nutrients—particularly short-chain fatty acids (SCFAs) [30]. Among the nutrients than can exert a central role in GM modulation, indigestible carbohydrates are of major interest as functional food ingredients with health benefits [31], as substrates to produce SCFAs. 

## 3. Chitin Content in Foods and Chitin Derivatives

Chitin is a polysaccharide composed of *N*-acetyl-2-amino-2-deoxyglucose (GlcNAc) units linked by β-(1→4) bonds. Chitin is the main fibrous compound in arthropod exoskeletons, mollusk radula, cephalopod endoskeletons, fungal cell walls, and fish and lissamphibian scales [32]. The largest source of chitin globally is suggested to be zooplankton cuticles, with an estimated 379 million tons of Antarctic krill available worldwide [32]. However, fishing these tiny organisms is not commercially viable; and subsequently, considered as shellfish industry waste, such as shrimp, crab and lobster shells with a chitin content of 8–40% are the main source of chitin [30,31]. Fungi provide an alternative source of chitin and, despite having a lower chitin content than crustaceans (10–26% as a chitin-β-(1,3/1,6) glucan complex, GC), are attracting increasing scientific and food industry interest [33,34].

The chitin content can widely vary between different sources, ranging from 16–23% in lobster shells, 25–30% in crab shells and 34–49% in krill shells to 18–38% in cockroach cuticles, 22–64% in butterfly cuticles, 20–44% in silkworm, 8–43% in mushrooms cell walls, 8–27% in mold cell walls and 1–3% in yeast cell walls [33]. Whereas crustacean exoskeletons are not usually employed by the food industry and are considered waste [35], and chitin from fungi is also often extracted from residues [36], chitin from insects can be ingested together with other nutrients because they are usually consumed as whole insects or parts of whole edible insects including ingredients derived from them such as meals/flours [4]. As consumer acceptance in developed countries remains one of the barriers to their use as an entire food [37], the inclusion of insects as flours or other forms of food, for example, cookies, energy bars, hamburgers and sandwich spreads, among others, is promising [38]. Diverse studies have revealed that males are more willing to adopt insects as a protein substitute than females as they are, on average, described as more adventurous eaters [37]. Similarly, younger consumers are less reluctant to eat insects than older consumers [37]. Because of their high chitin content, which represents at least 10% of all dry insects, insects can be a good source of DF in the human diet [4]. In addition to their chitin content, diets containing insects offer other important benefits for both animals and humans. Thus, for the case of animal feed, it was previously reported than insects are a good source of amino acids, fatty acids such as lauric acid, minerals, and most of the B group vitamins [39]. As other important benefit, due to their content in antimicrobial peptides and lauric acid, the inclusion of insects in feed contributed to a decreased need for antibiotics in animal rearing [39]. In the case of humans, insects were reported to be an excellent source of energy, fats, proteins, and minerals [4]. Insect-derived proteins were reported as of a higher biological value than those obtained from plant sources, with an essential amino acid score varying between 46–96%. The fatty acid profile of edible insects was reported as less saturated than those obtained from animal-origin foods, and depending on the species, with lower cholesterol levels, as well as containing plant sterols [40]. With respect to mineral supply, insects were reported as good sources of phosphorous, magnesium, manganese, copper, selenium, zinc, iron, and calcium [4].

Chitin is insoluble in water, but humans have digestive enzymes in their gastrointestinal tract that are capable of degrading chitin to some extent [2]. Chitinolytic enzymes (chitinase and chitobiase) break down the glycosidic bonds between GlcNAc units and degrade chitin into chitosan [2]. Lysozyme is also known to catalyze the deacetylation of 2-acetyl groups and the separation of glycosidic bonds between GlcNAc units from chitin, thus producing chitosan, a partially deacetylated byproduct of chitin degradation. The most beneficial advantage of chitosan is that it can be chemically modified into a wide variety of derivatives, that enhances some characteristics such as water-solubility [41]. Chitosan is a very useful and attractive biopolymer due to its diverse chemical structure. Its structural diversification can be seen by its molecular weight, which ranges from low (≤100 kDa) to high (≥300 kDa), as well as by its degree of deacetylation, which ranges from less than 60% in chitin to more than 60% in chitosan [41]. Hydrolyzed products of chitosan—*N*-acetyl-D-glucosamine oligomers (chitin oligosaccharide; NACOS) and D-glucosamine oligomers (chitosan oligosaccharide; COS), major degradation products of chitosan/chitin via chemical hydrolysis or enzymatic degradation involving deacetylation and depolymerization processes [3] with molecular weight ≤ 16 kDa [42,43]—are readily soluble in water because of their shorter chain lengths, and this solubility makes them especially useful for food industry purposes.

## 4. Effects of Chitin and Derivatives on Human Health

Additionally to the effects of chitin on the GM, where it has been reported to improve gastrointestinal health due to its prebiotic potential [2], when consumed in foods, chitin and its derivatives are functional DF that can reduce LDL-cholesterol levels in the blood [44,45]. Consumption of chitin has been shown to improve glucose intolerance, increase insulin secretion, relieve dyslipidemia, and protect intestinal integrity and the GM in mice treated with a high saturated fat diet [46]. Chitin, or its derivative, appears to have also antiviral, anticancer and antifungal activity, as well as antimicrobial properties and a bacteriostatic effect on the Gram-negative bacteria *Escherichia coli*, *Vibrio cholerae* and *Shigella dysenteriae* [47].

Chitosan, a naturally occurring bioactive polymer (a copolymer of *N*-acetyl-D-glucosamine and D-glucosamine) [48] is the most important chitin derivative, and has received increasing attention due to its specific biodegradability by colon bacteria, its well-documented biocompatibility, low toxicity and mutagenic properties [49]. Evidence has shown that chitosan possesses various biological activities, namely antioxidant, antitumor, anti-inflammatory, immunostimulant, wound healing, coadjutant (in aquatic animals), cholesterol-reducing, antibacterial and antifungal properties, and is useful as an active component of the diet for the loss of body fat. Chitosan has also been reported to contribute to decreased blood pressure, control of arthritis, treatment of diabetes mellitus and immunostimulation [21,50].

Previous studies have indicated that chitosan may inhibit the digestion and absorption of visceral fats and interfere with bile acid synthesis and lipid metabolism, demonstrating lipid-lowering effects. This indicates that oral administration of chitosan with *Ganoderma* polysaccharides improves lipid metabolism disorders [51]. Chitosan has been demonstrated to inhibit pancreatic lipase activity and bind to bile acids, resulting in reduced intestinal fat absorption and increased excretion of fecal fat. At the cellular level, chitosan can suppress adipocyte differentiation, triglyceride accumulation and expression of adipogenic markers, and has also demonstrated an increased ability to remove excess cholesterol and bile acids from tissues and carry them to the liver for excretion [52]. This is important considering that bile acids are potentially toxic [53]. In addition, the conversion of cholesterol to bile acids and their subsequent secretion into the bile is an important pathway for the elimination of cholesterol from the body [52]. Intervention with chitosan has been found to result in the lowest levels of total unsaturated fatty acids in fecal fats. Although the mechanisms involved in lipid metabolism following chitosan intake are not fully understood, enhanced absorption and/or binding function may play an alternative role [52].

Chitosan has also been used to chelate cholesterol in food, inhibiting its absorption in the human intestine [54]. It is important to note that chitosan particles and COS are non-allergenic bioactive nutrients [31,55,56]. It was recently demonstrated that chitosan nanoparticles can protect Caco-2 cells, a model of human enterocytes, from lipopolysaccharide (LPS)-induced cell membrane damage [57]. LPS, also known as lipoglycans and endotoxins, consist in large molecules formed by lipids and a polysaccharide composed of O-antigen, that are found in the outer membrane of Gram-negative bacteria [5]. According to previous reports, reducing the particle size and thus increasing the surface/volume ratio can improve the functional properties of biomolecules [48]. If chitin is degraded into chitosan and COS particles with a low enough molecular weight, they can be absorbed into the bloodstream and transferred to all organs and tissues, where they can have the beneficial effects mentioned above [2].

With respect to chitin derivatives, GC has been indicated as a dietary supplement, with the maximum rate of consumption set at 5 g per day for an average person [58]. Furthermore, GC consumption has been shown to reduce body weight gain in rats, thus pointing to GC as an interesting novel prebiotic for the prevention and treatment of obesity [59].

COS can modulate biological processes to protect the host from inflammation, immunity, obesity, microbial infection and diabetes. In vitro studies suggest that COS may reduce glucose transport in Caco-2 cells and promote glucose uptake by adipocyte cells, whereas in vivo studies show that COS may reduce fasting blood glucose in db/db mice and increase insulin secretion in type-1 diabetic mice [60]. The mechanism of action may be associated with inhibition of enzymes such as intestinal α-glucosidase and pancreatic α-amylase. In addition, it has been reported that COS may slightly affect the GM in cultures of human stool batches [61].

COS with a low molecular weight (<1000 Da) has been reported to significantly inhibit glucose absorption by the intestinal tract by suppressing pancreatic amylase and small-bowel glucosidase activity. In addition, COS increases insulin secretion by promoting the antioxidant capacity of the pancreas and exerts antidiabetic effects in db/db mice and in rats injected with streptozotocin [59]. In addition, COS intake increases serum insulin, which can be produced by beta cell proliferation [59].

## 5. Effect of Chitin and Its Derivatives on Gut Microbiota

Traditionally, both chitin and its derivatives are well known as inhibitors of the bacterial growth of pathogenic microorganisms such as *Salmonella* Typhimurium, enteropathogenic *E. coli* and *V. cholerae*, among others [23,61]. More recently, it has been reported that chitin ingestion, additional to inhibiting pathogen growth, does not show the same effect on some potentially beneficial bacteria, such as *Bifidobacterium* and *Lactobacillus* [4]. In two separate recent studies, chitin ingestion promoted the growth of beneficial bacteria in the GM. Stull et al. [62] reported that the presence of cricket chitin increased the multiplication of *Bifidobacterium animalis* by 5.7 times in the human adult GM. These effects were confirmed in a recent work that noted that the inhibition pathway of chitin against *Lactobacillus rhamnosus* (stimulation) and *E. coli* (inhibition), respectively, was different, the inhibitory activity against Gram-negative bacteria being higher than against Gram-positive [2].

In recent years, different research works have investigated the effect of whole insects, chitin and its derivatives on the GM in vitro, and in animal models and humans (Table 1). Some of these studies were performed in fish species [63,64]. 

Obviously, the results obtained in fish models are less applicable to humans because fish GM are very different to those of humans and terrestrial animals. Thus, it was reported that the predominant phylum in fish GM is Proteobacteria, followed by Fusobacteria and Firmicutes, whereas Bacteroidetes and Actinobacteria are often less numerous [48]. However, in the rainbow trout (*Oncorhynchus mykiss*) the most common phylum is Tenericutes, that is very scarce or non-existent in the human GM [56,64].

With respect to the effects of administering whole insects and insect flour on the GM, the results can be observed in Table 1. It shows works carried out in humans [62], on in vitro trials simulating the human gastrointestinal tract and using human feces [65], in laying hens or broilers [7,66,67,68], in fish species such as rainbow trout (*O. mykiss*) [63,64], zebrafish (*Danio rerio*) [69] and Siberian sturgeon (*Acipenser baerii*) [70], and in mice [71]. As can be seen in Table 1, the effects of insect administration or supplementation do not have a consistent effect on the GM. In most cases, there were a relevant number of beneficial effects, but in any case, it was found some harmful effects in GM composition were also found. These contradictory effects can be explained by the fact that both the insects and insect meal employed have a higher protein content that of chitin. A diet with a high protein content is well known to be dysbiotic, as usually occurs in the Western diet [10].

In vitro systems replicate the human GM but are less dynamic than the real human gastrointestinal environment. An in vitro trial simulating the human tract, investigating the administration of *Tenebrio molitor* flour, found an increase in Bacteroidaceae and Prevotellaceae, but not of harmful bacteria such as *Clostridium histolyticum*, Desulfovibrionales and Desulfuromonales. *Bacteroides* assists the host in degrading polysaccharides and contains genes codifying glucosidase enzymes [41], whereas *Prevotella* has the potential to participate in the metabolism and utilization of plant polysaccharides. Contrariwise, the relative abundance of Desulfovibrionales and Desulfuromonales is related to harmful effects and may contribute to the development of colorectal cancer [70], and *C. histolyticum* is potentially pathogenic in several species, including humans [65]. The beneficial effects were also reinforced in a study with an observed increase of SCFAs, very important for good colonic epithelial maintenance [65].

Clinical studies investigating prebiotic effects in humans have some issues with respect to ethical constraints, as well as limited sampling possibilities from the colon and limited measurements of in situ SCFA production [4]. In an in vivo clinical trial in humans [62], the intake of 25 g/day of *Gryllodes sigillatus* cricket powder provided contradictory results: an increase in probiotic species such as *Bifidobacterium animalis* and a decrease in other beneficial species such as *Lactobacillus reuteri*, as well as SCFAs. Both *Bifidobacterium* and *Lactobacillus* are well-known probiotic bacteria with a long history of safe use [5].

Diptera metabolites [68] showed beneficial effects regarding the management of *E. coli*-induced diarrhea in mice by modulating the immune system, antioxidants and GM composition. The intestinal microbiota imbalance was reversed, as shown by the increase in Firmicutes to Bacteroidetes ratio and *Clostridium* levels caused by *E. coli*-induced diarrhea.

With respect to trials carried out in mice, hens and chickens, the results showed benefits in metabolic parameters, such as an increase in the activity of glycolytic enzymes [69], a decrease in the triglyceride and cholesterol content in serum [7] and eggs [7,67], and an increase in the activity of α- and β-glucosidases and α-galactosidase [69], stimulating immunity and antioxidant capacity [71], but also harmful effects such as a decrease in villi mucin production [66] or laying frequency, feed intake and egg weight [67]. Production of mucin is important for correct maintenance of cecal epithelial integrity [66]. The changes in GM varied from an increase in richness [7,63,64] and diversity [7,64,66], or a decrease in diversity when used at higher concentrations [66], to increases in beneficial bacteria such as *Bacteroides* and *Prevotella* [65], *Bifidobacterium* [62,68], *Lactobacillus* [66] and *Ruminococcus* [66,71]. A reduction of *Clostridium* spp. was also found [71]. *Ruminococcus*, *Bacteroides*, *Bifidobacterium*, *Roseburia* and *Faecalibacterium* are considered primary degraders of complex polysaccharides, and their presence in abundance in the GM ensures good utilization and fermentation of DF [70].

However, harmful effects were also found, such as a decrease in *L. reuteri* [62], increase in *Helicobacter* [66], a cancer-related genus [68], decrease in *Bacteroides* and *Prevotella* [69], increase in the pathogen *Clostridium perfringens* [69] and increase in the phylum Proteobacteria [64], that contains pathogenic genera and species [21]. Thus, whole insects or their meal cannot be entirely considered as GM-enhancing foods. In any case, it should be considered that whole insects or insects meals have less dysbiotic effects on GM than other sources of animal protein [4]. Additionally, with respect to other chitin sources, such as crustaceans, insects are not overexploited than some crustaceans, specially shrimp, that are strongly exploited in all fishing areas, particularly in the Atlantic Ocean, where is considered fully exploited and in the Indian Ocean, where it were seems some signs of overexploitation [72].

The results obtained in fish species, although the changes at phylum levels are more difficult to extrapolate to humans due to the wide differences between the GM of fish and humans, showed beneficial effects in terms of greater GM diversity [63,64] or richness [64]. With respect to beneficial groups, an increase was observed in the butyrate-producing *Clostridium coccoides*–*Eubacterium rectale* cluster [71] and other lactic acid and butyrate producers [64]. Less consensus was obtained for other phyla such as Proteobacteria, that increased [64] or decreased [63] depending on the work, or Enterobacteriaceae, that also increased in Siberian sturgeon [70], whereas it decreased in zebrafish [69] supplemented in both cases with *Hermetia illucens*.

With respect to chitin derivatives, the results obtained can be found in Table 2. It includes works carried out in non-diabetic humans [73], in vitro trials [2,22,56,74,75], and in vivo trials using mice [46,57,76], rats [56,77,78], pigs [49,79], Syrian golden hamsters [80] and zebrafish [48]. Various types of chitin derivatives were included, such as chitosan [23,74,79], GC [56,73,76], NACOS [57], COS [2,23,46,50,75,77,78] and various different derivatives obtained from chitosan [48,49,80,81].

As can be seen in Table 2, the effects of chitin derivatives on both GM and metabolic effects are more favorable than those of whole insects or insect meal. Interestingly, an in vitro trial reported that the beneficial effects on GM of chitosan-carrying foods are limited to foods that possess a low protein content [23]. This could be a reasonable explanation for the better effects of chitin derivatives, due to the high protein content of both insects and insect-derived meals [4], and the dysbiotic effect of protein-rich diets [82,83,84].

Chitosan, the most simple chitin derivative, showed metabolic effects in pigs, such as a reduction in feed intake and body weight [79], and metagenomic evidence of promoting important metabolic pathways and vitamin synthesis [81], and improved serum lipidic profile, such as a decrease in triglycerides, cholesterol and aspartame aminotransferase [80], a marker of liver health [82]. Regarding its direct effect on the GM, chitosan addition was related to a decrease in Firmicutes [79,81], and increase in the counts of Bacteroidetes [81]. A higher ratio of Firmicutes to Bacteroidetes has previously been associated by some authors with a higher risk of obesity in humans [10,85]. Additionally, high proportions of Firmicutes have been associated with increased susceptibility to inflammation, infection, oxidative stress and insulin resistance [86,87]. Contrariwise, Bacteroidetes species such as *Bacteroides* and *Prevotella* possess strong peptidase activity and are associated with isovalerate and isobutyrate production [88]. At genus level, chitosan was found to increase *Prevotella* [81], decrease *Lactobacillus* [79,81] and increase *Bifidobacterium* [74,79], as well as increasing *Ruminococcus*, *Oscillibacter*, *Bifidobacterium*, *Prevotella*, *Alloprevotella* and *Paraprevotella* [80]. All the variations at genus level, with the exception of the decrease in *Lactobacillus*, are considered beneficial to health because they include SCFA producers, such as Prevotellaceae and *Oscillibacter*, that produce anti-inflammatory metabolites, which subsequently regulate proinflammatory immune cells [89,90,91].

With respect to GC, its administration in non-diabetic subjects reduced the serum content of oxidized low-density lipoproteins, a major risk in cardiovascular diseases, the leading mortality cause in Western countries [92]. GC administration also decreased the body weight of rats [56] and mice, accompanied by a decrease in fat mass [69].

Regarding the effect on GM composition, GC decreased the Firmicutes to Bacteroidetes ratio [56], and increased the amounts of lactic acid bacteria and SCFA-producing genera such as *Bifidobacterium* [56], *Bacteroides*–*Prevotella* spp., *C. coccoides*–*E. rectale* and *Roseburia* spp. All of these are genera that can grow in the presence of complex carbohydrates, and some of them such as *Roseburia* carry carbohydrate degradation genes, being producers of SCFAs [93]. *C. coccoides* produces metabolites, like SCFAs, secondary bile acids and indolepropionic acid, that play a probiotic role primarily through energizing intestinal epithelial cells, strengthening the intestinal barrier and interacting with the immune system [94].

NACOS has shown several pharmacological effects, including antimicrobial activity and protection against pathogen-induced infections [56]. The abundance of four beneficial bacteria, *Bifidobacterium*, *Lactobacillus*, *Akkermansia* and *Bacteroides*, were reduced in mice after a high-fat diet, which was significantly restored after supplementation with NACOS [56]. NACOS also clearly reduced the abundance in mice fed a high-fat diet of *Desulfovibrio* [56], a bacterial genus which is closely related to this type of diet [95] and is responsible for inflammation due to its lipid A structures of LPS [96]. *Desulfovibrio* can use hydrogen or organic compounds such as lactate and formate to reduce sulfate to generate hydrogen sulfide (H_2_S), which is toxic in nature and can have pathological consequences for the host [95].

Interestingly, it seems that treatment with NACOS significantly reduces mRNA levels of cytokines related to inflammation processes, including TNF-α, IL-6 and MCP-1, and also significantly decreases plasma concentrations of LPS in mice fed a high-fat diet [56]. NACOS also dramatically increased the abundance of *Bifidobacterium*, *Lactobacillus*, *Akkermansia* and *Bacteroides* in mice fed a high-fat diet. Evidence suggests that increased abundance of *Bifidobacterium*, *Lactobacillus*, *Akkermansia muciniphila* and *Bacteroides* has positive effects on intestinal integrity, glucose tolerance and attenuated obesity [97,98,99,100,101].

With respect to COS, some authors have hypothesized that its administration could provide better results than that of chitin or chitosan because of a more efficient digestion of chitin by AMCase in the gastrointestinal tract [2]. According to Mateos-Aparicio et al. [60], COS with many acetylated residues are more efficient in promoting the growth of beneficial *Lactobacillus* than deacetylated COS. These effects depend on the molecular weight and the degree of acetylation of COS [60]. The metabolic effect of COS showed beneficial effects on body weight maintenance and reduction, and glucose and insulin management, as well as decreasing inflammation-related markers [46,102]. When combined with resistant starch, COS administration decreased protein-fermentation markers such as H_2_S_2_, ammonia, phenols and indole as well as increasing the excretion of bile acids in feces, the thickness of the mucosal layer and SCFA production [78].

With respect to its effects on GM composition, COS did not have a significant effect on GM richness and diversity [46,75,77], but did stimulate the growth of *L. rhamnosus*, whereas chitin inhibited its growth [2], suggesting the above-mentioned more efficient digestion of COS than chitin in the gastrointestinal tract. It was also reported to have a favorable effect on the Firmicutes to Bacteroidetes ratio [46,75,77], decrease the phylum Proteobacteria [75], and reduce Lachnospiraceae NK4A136 group, *Alistipes*, *Helicobacter*, *Ruminococcus* and *Odoribacter*, while increasing Lachnospiraceae UCG 001 and *Akkermansia* [46]. These findings suggest benefits related to COS intake, as several studies indicate that the high abundance of *Alistipes* and decrease in the population of *Akkermansia* are closely related to the development of diabetes [95]. These results indicate the potential beneficial effects of COS on host health by reforming the structure of the GM.

Other chitosan derivatives such as low molecular weight chitosan [49,81] and chitosan–silver nanoparticles also showed a beneficial effect, both with respect to metabolic parameters such as goblet cell density and villi height [48], and to improving the metabolism of terpenoids and polyketides, digestive systems, cell growth and death, glycan biosynthesis and metabolism as well as the metabolism of cofactors and vitamins [81]. Effects on the GM were similar to those obtained from other chitin derivatives and consisted of benefits in the Firmicutes to Bacteroidetes ratio [49,81], a decrease in Proteobacteria [49], and increase in the genus *Prevotella* [81]. The only discordant element with respect to the beneficial effects was a decrease in *Lactobacillus* [49].

## 6. Conclusions

The decrease in terrestrial agriculture and disposable water is likely to increase the consumption of non-vegetable DF by humans in the near future, because its adequate consumption is essential to maintain human health. Chitin, as one of the most abundant biopolymers in nature, is an interesting candidate to partially substitute or complement cellulose-based DF, the main sources of chitin in nature being insects, crustaceans and fungi.

In view of the results obtained, the use of whole insects or meal, although showing beneficial effects on the modulation of GM composition in most cases, also showed some harmful effects on both GM composition and other metabolic parameters. This divergence could be explained by the high protein content of insects and meal that could contrast with the beneficial effects of chitin on GM composition and human health. Thus, as recent works have indicated, chitin derivatives such as chitosan only exert their potential prebiotic activity if carried by foods that possess a low protein content. However, chitin derivatives show better results in the modulation of GM, enhancing the growth of beneficial bacteria and inhibiting the growth of some potentially pathogenic bacteria. Additionally, chitin derivatives also show positive effects in terms of anti-inflammatory capacity, stimulation of the immune response, prevention of diabetes, and prevention and management of obesity. Thus, their potential as a source of DF should be investigated in more depth, to enhance the valorization of these products as human foods.

## Figures and Tables

**Table 1 foods-09-00782-t001:** Effects of administration of whole insects or insect flour on gut microbiota.

Type of Study	Insect	Dosage and Time of Administration	Significant Changes in Gut Microbiota	Significant Changes in Metabolites and Metabolic Effects	Reference
In vitro trial using 24 Lohmann brown classic laying hens	*Hermetia illucens* larvae meal	According to Marodo et al. (2017). Hens ingested around 1.02 g/day of chitin throughout the trial (21 weeks)	Increased GM diversity and richness, and increased proportions of *Oscillospira*, while proportions of *Fusobacterium* decreased	Increase in SCFA production, and lower triglyceride content in serum and cholesterol content in serum and egg yolks	[7]
In vivo using 20 healthy adults	*Gryllodes sigillatus* cricket powder	14 days of eating prepared study breakfast meals that included cricket powder (25 g/day) or control	*Bifidobacterium animalis* increased by a log fold-change of 5.7 on the cricket diet compared to the control diet. Probiotic species *Lactobacillus reuteri* and two other lactic acid-producing bacteria were decreased by 3- to 4-fold relative to the control after 2 weeks of cricket powder consumption	Acetate in the stool was reduced by an estimated 2.31 µM/g during the cricket diet. Similarly, cricket consumption was also associated with reduced propionate content	[61]
In vivo trial in rainbow trout (*Oncorhynchus mykiss*)	*H. illucens* insect meal	10%, 20% and 30% partial substitution of fish meal with insect meal for 12 weeks	Actinobacteria and Proteobacteria phyla were increased after inclusion of insect meal in the diet. An increase in richness, diversity and lactic acid- and butyrate-producing bacteria was also observed	Not provided	[64]
In vitro fermentation system using fresh fecal samples from 5 healthy donors	*Tenebrio molitor* flour	1% (*w*/*v*) of both digested and undigested *T. molitor* flour	Increase of Bacteroidaceae and Prevotellaceae, but not *Clostridium histolyticum*, Desulfovibrionales and Desulfuromonales	Ammonia production was, within concentration levels, considered not cytotoxic. Increased production of acetate and propionate, that are associated with promotion of satiety	[65]
In vivo trial using 256 broiler chickens	Partially defatted *H. illucens* larvae meal	5%, 10% or 15% meal in feed for 35 days	Increase in GM diversity in broilers supplemented with 5% and 10% *H. illucens* larvae meal. Increase in the proportions of Proteobacteria, *Lactobacillus* and *Ruminococcus*. Reduction in GM diversity and increase in *Bacteroides*, *Roseburia* and *Helicobacter* in broilers supplemented with 15% *H. illucens* larvae meal	Decrease in villi mucin production in broilers supplemented with 10% and 15% *H. illucens* larvae meal	[66]
In vivo trial using 104 Lohmann Brown classic laying hens	*H. illucens* larvae meal addition as total replacement of soybean meal	24 to 45 weeks	Not provided	Better feed conversion ratio but lower lay percentage, feed intake and egg weight. Lower cholesterol and triglycerides in eggs and higher calcium levels in blood were obtained in hens supplemented with *H. illucens* larvae meal	[67]
In vivo using 600 Roos 308 1-day-old broilers	*T. molitor* and *Zophobas morio* larvae	Feed enriched with insect meal according to the following experimental system: 0.2% *T. molitor*, 0.2% *Z. morio*, 0.3% *T. molitor* or 0.3% *Z. morio* for 35 days	Dietary insects significantly decreased the cecal counts of *Bacteroides*–*Prevotella* cluster. *Clostridium perfringens* counts were increased in the broiler chickens subjected to the 0.3% *Z. morio* treatment. The addition of *Z. morio* resulted in an increase of the relative abundance of Actinobacteria, including the family Bifidobacteriaceae, and the addition of *T. molitor* resulted in a significant increase of the relative abundance of family Ruminococcaceae	Addition of *Z. morio* (0.2%) increased the activity of glucosidases and α-galactosidase	[68]
In vivo using zebrafish (*Danio rerio*)	*H. illucens*	Two different *H. illucens* groups were reared on coffee byproducts or a mixture of vegetables for 6 months	Enterobacteriaceae counts in samples from fish fed both types of feed containing *H. illucens* were lower than those in controls	Not provided	[69]
In vivo using 180 juvenile Siberian sturgeon (*Acipenser baerii*)	*H. illucens* and *T. molitor* larvae meal	Diets were prepared by replacing fishmeal in the control diet with 15% *H. illucens* larvae meal and 15% *T. molitor* larvae meal. The daily feed rations varied from 1.8% to 1.4% for 60 days	The *H. illucens* diet increased *Clostridium leptum* subgroup, Enterobacteriaceae, *Clostridium coccoides*–*Eubacterium rectale* cluster, *Aeromonas* spp., *Bacillus* spp., *Carnobacterium* spp., *Enterococcus* spp. and *Lactobacillus* spp.The *T. molitor* diet decreased the total number of bacteria, but did not significantly affect the *Clostridium leptum* subgroup, Enterobacteriaceae, *Aeromonas* spp. or *Lactobacillus* spp.	The *H. illucens* diet caused modified intestinal histomorphology by reducing mucosal thickness and increasing muscle layer thickness	[70]
In vivo using 42 specific-pathogen-free mice	3-week-old stable fly larvae	Normal control group (*n* = 7); *Escherichia coli* control group (*n* = 7); ciprofloxacin (0.13 mg/kg; *n* = 7), 2 mg/kg metabolites of stable fly (*n* = 7), 4 mg/kg metabolites of stable fly (*n* = 7) and 8 mg/kg metabolites of stable fly (*n* = 7) groups. All groups but controls were injected intraperitoneally with *E**. coli* (0.3 mL of 2.50 × 10^11^ CFU/mL) once a day for 3 days, respectively, to induce diarrhea	Cecal microbial sequencing showed a significant difference in the prevalence of Firmicutes, *Clostridium*, *Bacteroidetes* and *Alistipes* in the metabolites of stable fly-treated groups, suggesting that this treatment could be used to manage diarrhea in mice	Metabolites derived from stable fly are rich in amino acids that may affect intestinal health by regulating intestinal immunity, antioxidant capacity and microbial population in mice with diarrhea	[71]

CFU: Colony formit units; GM: Gut microbiota; SCFA: Short Chain Fatty acids.

**Table 2 foods-09-00782-t002:** Effect of chitin derivatives on the gut microbiota different species.

Type of Study	Insect/Crustacean, Dosage and Time of Exposure	Characterization	Significant Changes in Gut Microbiota	Significant Changes in Metabolites and Metabolic Effects	Reference
In vitro determination of bacteria grown in tryptone soy broth comparing *E. coli* vs. *Lactobacillus rhamnosus*	Chitin was tested at concentrations of 5 and 1 g/L, whereas COS was tested at concentrations of 5, 1, and 0.5 g/L	COS powder ≤ 1.5 kDa and degree of deacetylation ≥ 90%. Chitin was prepared at a concentration in the range 1–5 g/L. COS was prepared in the range 0.5–5 g/L	COS reduced the growth of *E. coli*, whereas chitin totally inhibited *E. coli* growth. COS stimulated the growth of *L. rhamnosus*, whereas chitin inhibited its growth	Not provided	[2]
In vitro determination of minimal inhibitory concentrations against *E. coli* and *Staphylococcus aureus* in broth, milk and apple juice	Minimal inhibitory concentrations were tested in the range 0–0.6% *w*/*v*. 0.5% *w*/*v* chitosan and COS in milk and apple juice	Chitosan with average molecular weights of 628, 591 and 107 kDa and a degree of deacetylation in the range 80–85%. COS with molecular weight of < 5 and < 3 kDa and a degree of deacetylation in the range 80–85%	COS showed higher antibacterial activity than chitosan against *E. coli*, whereas chitosan showed higher antibacterial activity than COS in *S. aureus*. The use of chitosan in foods will be limited to foods that possess a low protein content	Not provided	[23]
In vivo trial using 24 C57BL/6J mice	1 mg/mL in water, about 200 mg/kg/day for 3 months	COS < 1 kDa with deacetylation degree of 88%	Significant decrease in Firmicutes phylum and increase in Bacteroidetes phylum in dp/dp mice. At genus level, markedly reduced Lachnospiraceae NK4A136 group, *Alistipes*, *Helicobacter*, *Ruminococcus* and *Odoribacter*, while Lachnospiraceae UCG 001 and *Akkermansia* increased	Lower fasting glucose, better insulin tolerance. Reduced weight of white fat tissue. Significant decrease in mRNA levels of inflammation markers such as TNF-α, MCP-1 and macrophage biomarker CD11c	[46]
In vivo trial using adult zebrafish (*Danio rerio*)	2% of zebrafish diet for 60 days	Chitosan–silver nanocomposites	Increase of Bacteroidetes, Fusobacteria and unassigned other phylum, whereas Proteobacteria decreased	Increase in goblet cell density and in villi height. Genes of IL-6 and 12 showed significantly higher regulation, whereas mucin-encoding genes, such as Muc 5.1 and Muc 2.1 showed upregulation in treated fish	[48]
In vivo trial using 144 piglets (Duroc × landrace × Yorkshire)	100, 200 or 400 mg/kg in feed for 28 days	Chitosan nanoparticles with a particle size of about 50 nm, average molecular weight of 220 kDa and degree of deacetylation of 95%	Increase in GM diversity and *Bacteroidetes*, Prevotellaceae and *Ruminococcus* while Firmicutes and Clostridiaceae family decreased	Improvement in growth performance. Improvement in immunoglobulin IgA, IgG, C3 and C4. Decrease in plasma cortisol, PEG2, IL-6 and IL-1ß	[49]
In vitro trial investigating effects on growth of 100 *Bifidobacterium* and in vivo trial in 24 Groningen rats	In vitro trial at 0.5% GC *w*/*v* concentration. Rat diet was supplemented with 10% GC (*w*/*w*)	GC was obtained from *A. niger* by a tyndallization procedure	Increase of *Bifidobacterium adolescentis* and *B. longum*. Decrease in Firmicutes to Bacteroidetes ratio and improved colonization efficiency of *B. breve*	Decrease in body weight gain with respect to controls	[56]
In vivo trial using 20 C57BL/6J mice	1 mg/mL chitin oligosaccharide (NACOS) in drinking water (about 200 mg/kg/day) for 5 months in a high-fat diet	NACOS with a polymerization degree 2–6	Increase of *Lactobacillus*, *Bifidobacterium*, *Akkermansia* and *Bacteroides* whereas *Desulfovibrio* and Firmicutes to Bacteroidetes ratio decreased	Decrease in mRNA of cytokines, including TNF-α, IL-6, MCP-1 and LPS in serum. Improved bacterial motility, oxidative stress, energy metabolism and inflammation process	[57]
In vivo using 130 subjects free of diabetes mellitus	Participants were randomly assigned to receive chitin–glucan (GC) (4.5 g/day; *n* = 33), GC (1.5 g/day; *n* = 32), GC (1.5 g/day) plus olive oil extract (135 mg/day; *n* = 30) or matching placebo (*n* = 35) for 6 weeks	GC derived from *Aspergillus niger* mycelium	Not provided	Administration of 4.5 g/day GC for 6 weeks significantly reduced oxidized low-density lipoprotein. At the end of the study, GC was associated with lower LDL-C levels, although this difference was statistically significant only for the GC 1.5 g/day group	[73]
In vitro trial using trypticase phyton yeast inoculated with different *Bifidobacterium* strains	0.025%, 0.1% and 0.5% low-molar-mass chitosan, chitosan succinate; chitosan glutamate and 0.1% and 0.5% COS in anerobic trypticase phyton yeast medium	Chitosan molecular weight 75 kDa; degree of deacetylation 83%, prepared by enzyme hydrolysis to obtain different fractions	Both chitosan and all derivatives inhibited *Bifidobacterium* growth	Not provided	[74]
In vitro fermentation using fresh feces of C57BL/6J mice	1 g/L of COS in drinking water, about 200 mg/kg/day for 5 months	COS with deacetylation degree over 95% and average molecular weight < 1 kDa	Increase of Bacteroidetes and Verrucomicrobia phyla whereas Proteobacteria and Firmicutes phyla decreased	Increase in colonic H_2_, acetate and butyrate	[75]
In vivo using 24 C57BL/6J mice	GC (10% *w*/*w*). Food intake was recorded, taking into account spillage, twice a week for 4 weeks	GC was derived from the cell walls of the mycelium of *A. niger*	GC supplementation increased the quantities of *Bacteroides*–*Prevotella* spp., whereas the *Clostridium coccoides*–*Eubacterium rectale* cluster group and *Roseburia* spp. were completely restored after GC treatment. Bifidobacteria in the high-fat GC-fed mice were higher than in the high fat-fed mice or control mice	GC decreased body weight gain by about 28% as compared to high-fat diet. This effect was accompanied by lower fat mass development. Consumption of GC showed potential beneficial effects with respect to the development of obesity and associated metabolic disorders such as diabetes and hepatic steatosis	[76]
In vivo trial using 40 male Sprague-Dawley rats	COS (0.3 g/day), resistant starch (1.2 g/day) and COS combined with resistant starch (1.5 g/day) slurried with drinking water for 6 weeks	COS with an average molecular weight about 5 kDa and a deacetylation degree of 83%	COS increased Bacteroidetes and decreased Firmicutes. COS combined with resistant starch *Blautia* and *Allobacterium*	COS combined with resistant starch decreased protein-fermentation markers such as H_2_S_2_, ammonia, phenols and indole. It also increased excretion of bile acids in feces, the thickness of the mucosal layer and SCFA production	[77]
In vivo using 12 Wistar rats	Control group received pellets with commercial diet ST-1. Treated group pellets had chitosan or COS added at a final concentration of 10 g/kg in feed mixture) for 4 weeks	COS obtained by cellulase hydrolysis of chitosan from *A. niger*	Increase of total bacterial population in the group of *Bacteroides*–*Prevotella* and the *Clostridium leptum* subgroup was found in response to chitosan intake. Chitosan intake also reduced Enterobacteriaceae and *Lactobacillus* group bacteria. COS intake influenced *Bacteroides*–*Prevotella* group and Enterobacteriaceae bacteria in the same way	Not provided	[78]
In vivo trial using 40 pigs	Basal diet plus 1000 µg/kg chitosan for 63 days	Chitosan obtained from prawn (*Nephrops norvegicus*)	Chitosan supplementation decreased Firmicutes in the colon and decreased *Lactobacillus* spp. in both the cecum and colon, while *Bifidobacterium* increased in the cecum	Reduced feed intake and body weight in pigs	[79]
In vivo trial using 24 Syrian golden hamsters with dyslipidemia previously induced with high-fat diet	150 mg/kg/day for 8 weeks	Chitosan with degree of diacylation higher than 85% combined to *Ganoderma* polysaccharides at 1:1 ratio	Increase of *Ruminococcus*, *Oscillibacter*, *Bifidobacterium*, *Prevotella*, *Alloprevotella* and *Paraprevotella*	Triglycerides, total cholesterol, low-density lipoprotein cholesterol and aspartate aminotransferase were reduced in the serum of hamsters fed chitosan-added diet	[80]
In vivo trial using 60 pigs	Basal diet with 50 g/Tm added for 28 days	Low molecular weight chitosan	Increase in Bacteroidetes, decrease in Firmicutes. Increase in *Prevotella* but decrease in *Lactobacillus*	Chitosan supplementation improved metabolic pathways including energy metabolism, metabolism of terpenoids and polyketides, digestive systems, cell growth and death, glycan biosynthesis and metabolism as well as metabolism of cofactors and vitamins	[81]

GC: Chitin-glucan; IL-1: Interleukin 1; IL-6: Interleukin 6; IgA: Inmunoglobulin A; IgG: Inmunoglobulin G; LDL-C: low-density lipoprotein cholesterol; LPS: lipopolysaccharide; MCP-1: monocyte chemoattractant protein-1; NACOS: chitin oligosaccharide; mRNA: messenger RNA; TNF-α: tumor necrosis factor alpha; SCFA: Short chain fatty acids; PEG2: Prostaglandin E-2.

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
