# Peer review of "Animal-Origin Prebiotics Based on Chitin: An Alternative for the Future? A Critical Review"

_foods, 2020, doi:10.3390/foods9060782_

Round 1

Reviewer 1 Report

General comments:

The authors describes beneficially potential of whole insects, crustaceans and polisaccharides obtained from animal sources (chitin and derivates) to be included in human diet as a source of complex polysaccharides. The human gut microbiota is a topic that is receiving high attention by the scientific community and this review article is interesting with important conclusions in this topic for human food chain. The review article seems to be carried out well designed and reported. There are parts in manuscript where English could be more improved. This paper could be suitable for publication while some improvements suggestions below have been made.

1)        Insects are promising feedstuffs for animal feeds and human foods as they contain not only chitin but also particular compounds that seem to be able to modulate microbiota like lauric acid, and anti-microbial peptides provided by insects. The authors should also must indicate beneficial effects of theese substances. Please check this reference: Can diets containing insects promote animal health?’ – L. Gasco et al., 2018, Journal of Insects as Food and Feed.

2)   The authors explain that whole insects cannot be entirely considered as microbiota enhancing foods and they promote crustaceans as one of main source of chitin in nature. Many authors indicate that world shrimp and also crustaceans resources are strongly exploited in all fishing areas (M.G. Bondad-Reantaso et al./Journal of Invertebrate Pathology 110 (2012) 158–165; Lai et al./ Ocean & Coastal Management Volume 187 (2020), 105-104). The authors should explain their opinion about.

Specific comments:

Lines 33-34, please delete sentence;

Line 50, add reference;

Line 65 specify FAO and add year;

Lines 89-91, please explain better;

Line 98, “animal-derived DF on the human” only part of;

Line 199, add other references;

Line 221, “This is important considering that bile acids are potentially toxic” add reference.

Author Response

Rebuttal letter manuscript foods-804171

Many thanks for the careful revision that been done to our article.  All the comments and suggestion highlighted by reviewers have been considered and corrected or incorporated in the revised version of the manuscript. We thank to the Reviewers because their comments allowed to strongly improve our manuscript.

With respect to the comments from Reviewer 1:

With respect to the comments about: The authors describe beneficially potential of whole insects, crustaceans and polysaccharides obtained from animal sources (chitin and derivates) to be included in human diet as a source of complex polysaccharides.

The human gut microbiota is a topic that is receiving high attention by the scientific community and this review article is interesting with important conclusions in this topic for human food chain. The review article seems to be carried out well designed and reported. There are parts in manuscript where English could be more improved.

Response: We greatly appreciate the constructive comments from the Reviewer. We value your opinion very much, as well as your comments that allowed to improve the original version of the manuscript.

Obviously, none of us are native English speaker, and consequently it is probable that we may have corrected some writing errors. However, prior to sending the manuscript, we send it to a professional editing service. We provide a copy of the certificate issued by that editing service.

In any case, we are willing to re-contract the services of another proofreading service if the reviewer or editor deems it necessary

With respect to the comments about: Insects are promising feedstuffs for animal feeds and human foods as they contain not only chitin but also particular compounds that seem to be able to modulate microbiota like lauric acid, and anti-microbial peptides provided by insects. The authors should also must indicate beneficial effects of these substances. Please check this reference: Can diets containing insects promote animal health?’ – L. Gasco et al., 2018, Journal of Insects as Food and Feed.

Thank very much for your comment. In fact, other beneficial compounds can be obtained by insects additionally to chitin. According to the suggestion from the Reviewer, in the revised version of the manuscript, it was included the following paragraph:

“Additionally than their chitin content, diets containing insects contains important benefits for both animals and humans. Thus, for the case of animal feed, it was previously reported than insects are a good source of amino acids, fatty acids such as lauric acid, minerals, and most B vitamins [39]. As other important benefit, due their content in antimicrobial peptides and lauric acid, the inclusion of insects in feed contributed to decrease the need of antibiotics in animal rearing [39]. In the case of humans, insects were reported to an excellent source of energy, fats, proteins, and minerals [4]. The insect-derived proteins were reported as of a higher biological value than those obtained from plant sources, with an essential amino acid score varying between 46-96%. The fatty acid profile of edible insects was reported as less saturated than those obtained from animal-origin foods, and depending on species, with lower cholesterol levels, as well as containing plant sterols [40]. With respect to mineral supply, insects were reported as good sources of phosphorous, magnesium, manganese, cooper, selenium, zinc, iron, and calcium [4].”

As well as the following references were included in the references list:

Gasco, L.; Finke, M.; van Huis, A. Can diets containing insects promote animal health? J. Insects as Food Feed. 2018, 4(1), 1-4.

Sabolová, M.; Adámková, A.; Kourimská, I.; Chrpová, D.; Pánek, J. Minor lipophilic compounds in edible insects. Potravinarstvo Slovak J. Food Sci. 2016, 10, 400-406.

With respect to the comments about: The authors explain that whole insects cannot be entirely considered as microbiota enhancing foods and they promote crustaceans as one of main source of chitin in nature. Many authors indicate that world shrimp and also crustaceans resources are strongly exploited in all fishing areas (M.G. Bondad-Reantaso et al./Journal of Invertebrate Pathology 110 (2012) 158–165; Lai et al./ Ocean & Coastal Management Volume 187 (2020), 105-104). The authors should explain their opinion about.

Response: Thank you very much for your comment. Please, note that we do not recommend a specific source of chitin as best than other sources. In fact, when insects as consumed whole on in meals, they contain all proteins from non-cuticle parts of insects. Consequently, because insects are a very good source of proteins, proteins exert a dysbiosis effect. We have not compared the effects of chitin obtained from insects with chitin obtained from crustaceans, and we do not believe that the origin of chitin has a relevant effect on its effects on the intestinal microbiota. In any case, your comments are interesting, and we consider it important to stress in the manuscript that unlike crustaceans, insects are by no means overexploited and may be a more sustainable source of chitin. Thus, in the revised version of the manuscript, according to your suggestion, it was added the following paragraph:

“In any case, it should be considered that whole insects or insects meals have less dysbiotic effects on GM than other sources of animal protein [4]. Additionally, with respect to other chitin sources, such as crustaceans, insects are not overexploited than some crustaceans, specially shrimp, that are strongly exploited in all fishing areas, particularly in the Atlantic Ocean, where is considered fully exploited and in the Indian Ocean, where it were seems some signs of overexploitation [72].”

Therefore, the following reference was added to the references list:

Bondad-Reantaso, M.G.; Subassinghe, R.P.; Josupeit, H.; Cai, J.; Zhou, X. The role of crustacean fisheries and aquaculture in global security: past, present and future. J. Invertebr. Pathol. 2012, 110, 158-165.

With respect to the comments about: Lines 33-34, please delete sentence.

According to the suggestion from the Reviewer, lines 33-34 were deleted in the revised version of the manuscript.

With respect to the comments about: Line 50, add reference.

According to the suggestion from the Reviewer, two references reinforcing the reference were added.

With respect to the comments about: Line 65 specify FAO and add year.

According to the suggestion from the Reviewer, year of the document was added to the main text. FAO was previously defined as “Food and Agriculture Organization of the United Nations” in line 40.

With respect to the comments about: Lines 89-91, please explain better.

According to the suggestion from the Reviewer, the text was changed to “The human gastrointestinal tract harbors 10–100 times the number of total eukaryotic cells of the human body. Their counts are especially high in the distal part of the colon, reaching 1011-1012 bacteria per gram [10]. Consequently, distal part of the colon contents one of the highest densities of bacteria on the earth [10].” For a better understanding

With respect to the comments about: Line 98, “animal-derived DF on the human” only part of.

According to the suggestion from the Reviewer, it was specified in the revised version of the manuscript that only a part (some) animal-derived DF were investigated.

With respect to the comments about: Line 199, add other references;

According to the suggestion from the Reviewer, it were added two new references reinforcing the cited sentence.

With respect to the comments about: Line 221, “This is important considering that bile acids are potentially toxic” add reference.

According to the suggestion from the Reviewer, it was included a new reference describing the potentially toxicity of bile acids.

Reviewer 2 Report

It is a real good review article, has a scientific value and novelty, is written legibly and all information are presented in a logical way. It describes an actual state of the art in this field. I have read the article with interest and pleasure.

Authors described chitin, and their derivatives as chitosan on the impact on human gut microbiota (GM). Chitin can be obtained from non-vegetable sources, the best being insects, crustacean exoskeletons and fungi. The review offers a broad perspective of the current knowledge surrounding the impacts of chitin and its derived polysaccharides on the human GM. The authors described both the pros and cons of a diet rich in chitin and chitosan on the human health and GM. I think, authors should consider changing the article title to ,,Animal-origin prebiotics based on chitin: An alternative for the future? A critical review" to underline the information contained in their article.

Author Response

With respect to the comments from the Reviewer 2:

With respect to the comments about: It is a real good review article, has a scientific value and novelty, is written legibly and all information are presented in a logical way.

It describes an actual state of the art in this field. I have read the article with interest and pleasure.

Authors described chitin, and their derivatives as chitosan on the impact on human gut microbiota (GM). Chitin can be obtained from non-vegetable sources, the best being insects, crustacean exoskeletons and fungi.

The review offers a broad perspective of the current knowledge surrounding the impacts of chitin and its derived polysaccharides on the human GM.

Response: We greatly appreciate the constructive comments from the Reviewer. We value your opinion very much, as well as your comments that allowed to improve the original version of the manuscript.

With respect to the comments about: The authors described both the pros and cons of a diet rich in chitin and chitosan on the human health and GM. I think, authors should consider changing the article title to “Animal-origin prebiotics based on chitin: An alternative for the future? A critical review" to underline the information contained in their article.

Response. According to the suggestion from the Reviewer, we changed the title of the manuscript to the proposed by the reviewer.